# Behavioral, Neurochemical and Developmental Effects of Chronic Oral Methylphenidate: A Review

**DOI:** 10.3390/jpm13040574

**Published:** 2023-03-23

**Authors:** Daniela Senior, Rania Ahmed, Eliz Arnavut, Alexandra Carvalho, Wen Xuan Lee, Kenneth Blum, David E. Komatsu, Michael Hadjiargyrou, Rajendra D. Badgaiyan, Panayotis K. Thanos

**Affiliations:** 1Behavioral Neuropharmacology & Neuroimaging Laboratory on Addictions (BNNLA), Clinical Research Institute on Addictions, Department of Pharmacology and Toxicology, Jacobs School of Medicine and Biomedical Sciences, University at Buffalo, Buffalo, NY 14203, USA; 2Division of Addiction Research & Education, Center for Psychiatry, Medicine & Primary Care (Office of the Provost), Western University Health Sciences, Pomona, CA 91766, USA; 3Department of Orthopaedics and Rehabilitation, Stony Brook University, New York, NY 11794, USA; 4Department of Biological and Chemical Sciences, New York Institute of Technology, Old Westbury, NY 11568, USA; 5Department of Psychiatry, UT Health San Antonio, San Antonio, TX 78229, USA

**Keywords:** methylphenidate, ADHD, Ritalin, psychostimulant, dopamine, animal model, neurochemistry, behavior

## Abstract

The majority of animal studies on methylphenidate (MP) use intraperitoneal (IP) injections, subcutaneous (SC) injections, or the oral gavage route of administration. While all these methods allow for delivery of MP, it is the oral route that is clinically relevant. IP injections commonly deliver an immediate and maximum dose of MP due to their quick absorption. This quick-localized effect can give timely results but will only display a small window of the psychostimulant’s effects on the animal model. On the opposite side of the spectrum, a SC injection does not accurately represent the pathophysiology of an oral exposure because the metabolic rate of the drug would be much slower. The oral-gavage method, while providing an oral route, possesses some adverse effects such as potential animal injury and can be stressful to the animal compared to voluntary drinking. It is thus important to allow the animal to have free consumption of MP, and drinking it to more accurately mirror human treatment. The use of a two-bottle drinking method allows for this. Rodents typically have a faster metabolism than humans, which means this needs to be considered when administering MP orally while reaching target pharmacokinetic levels in plasma. With this oral two-bottle approach, the pathophysiological effects of MP on development, behavior, neurochemistry and brain function can be studied. The present review summarizes these effects of oral MP which have important implications in medicine.

## 1. Introduction

### 1.1. History of Methylphenidate (MP)

Methylphenidate (MP), more commonly known as Ritalin, was created in 1944 by Panizzon [1]. In 1955, MP was used to treated psychological disorders such as depression, chronic fatigue, and schizophrenia [1]. However, in 1961, MP was approved for use, and demonstrated the most efficacy, on patients diagnosed with hyperactivity [1]. MP is now available in two forms: immediate release and sustained release [2]. Studies have shown that immediate-release MP is more likely to cause stimulant-like side effects, such as tachycardia and sweating, whereas sustained release does not [3]. The use of MP has doubled in the last decade, reaching its peak in 2012; however, it has been steadily declining since [4]. Two-thirds of children, 10% of adolescent boys, and between 1.5% and 31% of college students diagnosed with attention-deficit/hyperactivity disorder (ADHD) are treated with psychostimulants such as MP [5]. In 2003, a study concluded that approximately 4% of undergraduate students use MP illicitly [6]. Illicit use of MP can lead to similar effects on the body as cocaine; both drugs are psychostimulants and act akin. Illegal use of MP induces a quick and large dopamine (DA) release, creating a euphoric experience for the consumer [7]. On the contrary, therapeutic use of MP provides a constant DA release [7]. Typical dosages of MP range from 10 to 60 mg/kg [2].

### 1.2. MP Prescription and Use in Humans

MP commonly used to treat ADHD has shown progression throughout its existence. Common doses of MP in adults are between 20 and 30 mg/kg and typically exceed 60 mg/kg [4]. Between the years of 2006 and 2016, MP use has increased from 7.9–20 tons, hitting its peak in 2012 at 19.4 tons [4]. Some clinical studies have shown that the abuse of MP increases in participants with drug dependence [8]. Drugs that can be absorbed rapidly (immediate release, IR) have shown association with drug abuse in comparison to those with a slower absorption rate (extended release, ER) [9]. The rate of absorption is highly influenced by the route of administration. Another factor that can lead to the misuse of MP is lack of sleep. In a previous study, it was shown that adult humans who averaged four hours of sleep voluntarily chose to consume 10 mg/kg of MP more often than those who averaged 6–8 h [10]. The dose and illicit use of MP have a proportional relationship. As the dose of MP increases, the probability of substance abuse increases as well [3]. As the human body develops throughout life, a lot of the effects can vary between ages, especially to the DAergic system. Studies have shown that after just four months of treatment, there are increases in cerebral blood flow to the DAergic system in children and these effects tend to be more permanent than if the drug was introduced during adulthood [11]. Studies have also demonstrated that chronic use of MP has long-lasting effects, whereas acute use does not [11]. Narcolepsy has also been successfully treated with MP by using specific dosing of MP, 9 mg/day, and slowly increasing over a year to 18 mg twice per day [12]. Following three months of this treatment, patients became asymptomatic [12]. MP was found to increase heart rate, helping with narcoleptic episodes [12]. In a case study in 1996, a 27-year-old male was treated with 10 mg MP BID (two times a day) and increased to 30 mg BID over the course of four months [13]. In this study, daytime sleepiness was reduced after one month of this treatment; MP was found to stimulate the central nervous system, which helped decrease sleepiness [13].

### 1.3. MP Off-Label Use and Abuse

Illicit use of MP has been an increasing concern among adolescents. The use and abuse among neurotypical people can have a diverse effect compared to those who are neurodivergent. Adolescents have been found to be the population that most frequently uses off-label MP [14]. Long-term recreational abuse of MP can lead to an alarming reduction in weight and even depressive episodes [15]. Due to MP being prescribed, unlike illegal psychostimulants such as cocaine, it is more readily available for consumption among the population. Aside from the direct effects that MP has on a neurotypical person, there is concern that it may lead to the consumption of stronger psychostimulants such as cocaine. If a tolerance to MP is established, the person who is consuming it off-label may seek a stronger drug to acquire the same euphoric effect. Approximately, one-quarter of college-aged illicit prescription stimulant users reported illicit use of MP; these students’ reasoning for the off-label use of MP was for enhancement in concentration [6]. Due to the augmentation in cognitive function, there is concern about the abuse of these drugs among students in higher levels of education, such as medical school [16]. 

## 2. Clinical: Effects of Oral MP on Behavior and Medicine

MP is widely prescribed for patients diagnosed with ADHD, helping to mediate behavior and attention deficits. MP treatment for ADHD leads to a remarkable enhancement in attention [17], improved gait [18], and improved motor function [19] without disrupting sleep [20] or neurophysiology [21]. It has been found that MP helps to alleviate symptoms of ADHD, such as hyperactivity [22]. Additionally, this drug has an efficacy of 70% when relieving symptoms compared to ADHD patients that were treated with a placebo [23]. MP has been found to regulate serotonin and melatonin levels, therefore balancing biological rhythms. This was found to help treat symptoms of ADHD [24]. Children with ADHD have been found to have difficulty in the morning. After a 12 h treatment with MP, these children were found to have an easier time getting ready as they had no functional impairment [25]. MP helps increase working memory; it was found that following two years of treatment, MP successfully improved memory, in comparison to the discontinuation group in which methylphenidate was gradually reduced to placebo, followed by complete placebo [26]. 

MP is also prescribed for the treatment of narcolepsy [13]. MP can aid in minimizing sleep episodes commonly found in narcolepsy [27]. Following a three-week treatment, MP was found to show improvements in narcolepsy patients [12]. Although modafinil is the drug of choice among adult patients with narcolepsy, it is not well tolerated among children; MP, however, was found to be a successful and well-tolerated treatment regimen among narcoleptic pediatric patients [28]. The success in using MP as a treatment for narcolepsy has been attributed to norepinephrine release [13].

## 3. Clinical: Effects of Oral MP on Brain Function and Neurochemistry

MP works by blocking the reuptake of neurotransmitters dopamine (DA) and norepinephrine (NE) back into the receptors, thus allowing for increased concentrations of these neurochemicals to encourage more effective binding to receptors [29,30]. It is believed that the increased DA and NE signaling may help regulate attention levels by affecting signaling, leading to a decrease in the spontaneous firing of neurons, and an increase in the signal to noise ratio, which may lead to elevated attention levels [29,31,32]. Unlike DA and NE, conflicting results have been reported regarding the influence of MP on the neurotransmitter serotonin; one study conducted by Kucenzki et al. found that, unlike amphetamines, MP does not have an effect on extracellular serotonin levels [33]. However, Daniali et al. reported higher levels of serotonin transporter (SERT) density in the medial frontal cortex (MFC) of adult rats after both short-term and long-term chronic exposure to MP [34]. 

A study conducted by Volkow et al. [29] analyzed brain glucose metabolism (BGluM) using positron-emission tomography (PET) on 23 healthy adults to find that the whole brain metabolism increases in individuals when working on a more labor-intensive cognitive task, such as a math test, compared to a simple task, such as looking at pictures; however, after oral MP (20 mg) intake, BGluM decreased while performing the cognitive task, implying more efficient use of the brain when focusing on the more mental labor-intensive task. Another PET study [35] on 50 healthy participants, half male and half female, found MP reduced the cost of mental labor and increased the choice of cognitive task over a leisurely task; they found this effect greatest in participants with the highest levels of striatal DA, indicating a relationship among MP, DA enhancement, and increased attention and cognitive efforts. Similar results have been found when testing ADHD individuals, rather than healthy non-ADHD participants. A study [7] analyzed 20 ADHD individuals who were evaluated before and after 12 months of oral MP treatment to find a reduction in impulsivity and hyperactivity with long-term treatment; a challenge dose of MP was administered and coupled with PET imaging technology to find a significant increase in DA in the ventral striatum of the brain, which was related to the reduction in symptoms.

These findings relating MP to the dopaminergic system led researchers [36] to wonder whether there are gender-based differences in the brain DA system that could affect sensitivity to stimulant medications. They used PET imaging to evaluate MP-based increases in DA in the striatum using different methods of MP administration in 95 healthy adults, 65 male and 30 female, where Cohort A received oral 60 mg MP and Cohort B received intravenous 0.5 mg/kg MP. These researchers found that females reported feeling increased levels of “drug effects” and demonstrated significantly higher DA release in the ventral striatum, but not the dorsal striatum, during both oral and intravenous MP administration compared to males. Researchers suggest that possible gender-specific increases in sensitivity specific to the DA system may be an underlying factor in gender differences seen in ADHD.

Questions regarding the effects of MP on brain structure have also been asked. Researchers [37] analyzed the effects of chronic MP use on brain structure in 131 adult patients with ADHD using MRI technology. Images were taken at baseline, after 3 months, and after 12 months of MP use. The study found that chronic MP use did not lead to any detectable cerebral loss in volume. Evidence from a review paper [38] of structural MRI studies indicate that long-term MP use may actually normalize structural brain changes in the white matter, anterior cingulate cortex, and cerebellum of children with ADHD.

The neurobiological effects through which MP works is still being explored. However, recent studies [39,40,41,42,43] found the use of MP to improve performance during cognitive tasks. A study in healthy men, and following the use of fMRI, found those subjects to have higher activation in the dorsal attention network (DAN) region of their brains, including the parietal and prefrontal cortex (PFC), and more deactivation in the default mode network (DMN) when compared to control groups. The authors suggest that MP, through the elevation of DA and NE signaling, alters activation in the DAN and DMN, ultimately impacting cognitive abilities [41].

Another imaging study [44] attempted to analyze the long-term effects of chronic MP use in children with ADHD, focusing on the neural networks related to executive functioning. Nine boys with ADHD were scanned while drug naive, then a year later after chronic MP treatment, and compared to controls who had never undergone treatment. They found no changes in brain activation patterns when comparing the children who had undergone treatment to those who had not.

A hallmark of ADHD is hyperactivity and researchers [45] conducted a study analyzing regional cerebral blood flow (rCBF) using PET to understand how chronic MP use changes resting brain metabolism and how these results correlate to behavioral changes in response to the drug. Scans were taken for 10 adults with ADHD while unmedicated and after three weeks of chronic MP use. The study concluded that chronic MP use increases rCBF in the cerebellar vermis and decreases it in the precentral gyrus and caudate nucleus, two areas noted for their role in motor function. Lower brain activity in these regions may correspond to decreased levels of hyperactivity from MP use.

## 4. Preclinical Models of MP Treatment

The use of animal models in clinical studies has been proven to be vital and indispensable in neuroscience and behavioral research. Laboratory rats and mice also provide ideal models for biomedical research and comparative medicine studies due to their similarities to humans in terms of anatomy and physiology [46]. The use of rodents in research also has economic and biological advantages. Rats and mice are small animals and require little space and resources to maintain. They also have shorter gestation times and produce larger numbers of offspring [46]. Due to their relatively larger brain sizes, rats are preferred to mice for brain surgery, imaging, and developmental studies [47]. Rats also have faster developmental stages when compared to humans, where one rat day is equivalent to around 27 human days [48]. This allows researchers to observe desired effects more rapidly than if the studies were conducted in humans. However, drug dose and route of administration are important factors to consider when designing animal studies used for neuropsychopharmacology research. Calculation of drug dosages needs to take the physiological and metabolic systems of rats into account. Almost all physiological and metabolic systems in a rat are faster than a human, including heart rate and respiratory rate [48]. Therefore, the doses administered must be adjusted to account for metabolic differences in each species when conducting research to obtain desired results for drug exposure and to prevent drug overdose.

Route of administration is another factor that is important to consider when creating and conducting animal studies. The method of exposure should be relatively similar between the animal model and what is seen in humans. Prior to the development of the two-bottle method of exposure, previous models for exposure to MP in rats have demonstrated to be unlike the exposure seen in humans. These models of MP exposure include intraperitoneal (IP) injections, subcutaneous (SC) injections, and oral exposure via oral gavage. Depending on the study, the selection of a particular route of administration and assessing the effective dose can affect the pharmacokinetics of the given substance [49,50]. Studies have demonstrated that the choice of route of administration can result in behavioral and neurochemical consequences associated with MP administration in rodents [51]. 

### 4.1. Intraperitoneal Injection of MP

IP injections are administered in the lower right abdominal quadrant of the animal away from the midline. They are frequently used in experiments to mimic a similar exposure method to oral exposure. IP injections will allow the drug to absorb more efficiently into the mesenteric vessels, in which the drug will likely undergo hepatic metabolism [50]. This route of absorption closely mirrors the route of absorption for oral exposure. However, limitations of this method include potential injury to the animal if the injection is performed incorrectly. By injecting too close to the surface, the drug is administered subcutaneously instead of intraperitoneally which can ultimately change the effects produced by the drug. Errors such as these can decrease the drug’s half-life and cause quick peak release of DA in the brain, which induces behavioral sensitization [14]. Another form of exposure that has been used to administer MP is through the SC route [52]. Dosages of MP administered to rodents are selected based closely on mirrored doses used by humans. By comparison, the process of performing IP injections in rats and mice only differ due to the size of the model [53]. Mice are much smaller than rats, therefore executing an IP injection is much easier. Studies indicate that doses of 2–5 mg/kg reflect the clinical use of MP, while 10–20 mg/kg dose emulates the “recreational” use of MP [54]. Doses of 2.5 mg/kg or greater of MP through IP injections are shown to produce an increase in locomotor activity whereas doses of 1 mg/kg or lower have no effect on locomotor activity [55]. Studies using IP injections have shown to be effective due to its fast absorption rate compared to other methods of administration. This is primarily because IP-administered pharmacological agents are exposed to a large surface area, close to that of the entire skin surface, which leads to rapid and efficient absorption (see Table 1) [56].

#### 4.1.1. Effects of Injected MP on Behavior

MP exposure using IP injections has allowed studies to achieve the desired plasma and brain levels of MP in rats that one would see in clinical studies [57]. The most commonly used IP injection dose is 2.0 mg/kg. However, studies that use 1.0 mg/kg MP have shown similar results when compared to studies that use 2 mg/kg of injected MP. Some studies assessed the long-term behavioral changes with chronic MP treatment in ADHD animal models. The study that used 1 mg/kg MP IP on ADHD rat models demonstrated a reduction in hyperactive behavior in MP-treated rats, linked to decreased levels of monoamines in the PFC [58]. Another study that used 2 mg/kg MP IP on ADHD rat models showed reduced self-administration and reinstatement of drug-seeking behavior when compared to wild-type rats. Results from this study suggested that reduced drug sensitization and tolerance after chronic MP exposure could be due to the neurobiological deficit of ADHD rat models [59]. IP-injected rats also demonstrated a significantly greater locomotor activity than oral gavage due to higher DA levels in the nucleus accumbens (NACC) from a study by Gerasimov et al. [51]. A study by Schmitz and colleagues [60] investigated the functional behavioral effects of chronic MP treatment via IP in juvenile rats and demonstrated that exploratory and object recognition memory can be impaired by chronic MP treatment due to increased neuroinflammation in the brain resulting from increased cytokines levels. Chronic MP exposure in juvenile rats has been shown to impair spatial and working memory when assigned to working memory tasks, which is conducive to previous studies that demonstrated behavioral changes associated with injected MP exposure such as depressive-like behaviors, anxiety-like behaviors, and decreased sensitivity to rewarding stimuli [57]. In contrast to chronic MP IP exposure, preadolescent exposure to MP in healthy rats is seen to cause beneficial complex behavioral adaptations, such as reduced sensitivity to a given reward, but also detrimental behavioral adaptations such as increased depressive-like behaviors and reduced habituation in a familiar environment [61]. Early exposure of low and high doses of MP is shown to induce cross-sensitization to amphetamine but not behavioral sensitization in adult rats, which suggests that early MP treatment might increase the risk for future drug abuse [62]. Another study on early MP exposure conducted by Crawford et al. [63] demonstrated an increase in the rewarding effects of cocaine in a self-administration (CSA) paradigm but not in cocaine-induced conditional place preference (CPP). These results suggest that early MP exposure is insufficient to induce long-term changes in neuronal reward system functioning. Pre-exposure to MP also enhances CSA with an increase in sensitivity to cocaine in adult rats when they were given MP pretreatment during adolescence [64]. 

#### 4.1.2. Effects of Injected MP on Brain Function and Neurochemistry

Due to the invasiveness of an IP injection, it is observed that MP levels in plasma and the brain are higher than those found in oral gavage exposure. MP exposure via IP exposure also increases DA levels in the brain at a faster rate with a much longer-lasting effect compared to oral gavage exposure [51]. IP injections are shown to be more suitable for chronic or repetitive drug treatments and are safer and well-tolerated in laboratory animals [56]. A few studies demonstrated that high-dose, chronic MP exposure via IP causes oxidative damage, inflammatory changes, and neurodegeneration in the brains of healthy young rats. These effects could be due to a dose-dependent increase in lipid peroxidation or due to an increased production of mitochondrial superoxide in the cerebellum, PFC, hippocampus, and striatum of healthy young rats [65,66]. Other than that, prominent DNA damage of striatal cells in both adult and young rats can be seen in chronic injected MP treatment. The cause of DNA damage is speculated to be due to DA oxidation, which leads to the generation of free radicals [67]. By contrast, in a more recent study, acute MP treatment is shown to have neuroprotective effects by reducing cell damage and decreasing apoptosis in brain tissues through the phosphorylation of the AMPK signaling pathway (See Table 1) [68]. A study done by Urban and colleagues [1] proposed that chronic MP treatment could affect prefrontal neurons in an age-dependent manner. The study demonstrated that chronic MP treatment enhances pyramidal activity in adult rats but reduces neuronal excitability and synaptic transmission of prefrontal neurons in juvenile rats. In a study conducted by Schmitz et al. [60], chronic MP IP injections of 2.0 mg/kg delivered once a day was shown to lead to a loss of astrocytes and neurons with increased cytokines and neurotrophin levels in the hippocampus of juvenile rats, which contributed to cognitive impairment due to increased levels of neuroinflammation.


jpm-13-00574-t001_Table 1Table 1Summary of behavioral and neurochemical effects on injected MP.MP ExposureBehavioral EffectsModel Used/ReferencesNeurochemical EffectsModel Used/ReferencesChronic*Decreased* hyperactive behavior *Decreased* self-administration and reinstatement of drug-seeking behavior *Decreased* drug sensitization and tolerance in exploratory and object recognition memory Impaired spatial and working memory results in *decreased* sensitivity to reward stimuli *Increased* locomotor activity compared to gavage administration *Increased* depressive and anxiety-like behaviorNaples high-excitability rats [58]Spotaneously hypertensive rats [59]Wistar rats [60]Wistar rats [57]Sprague-Dawley rats [51]Induces oxidative damage, inflammatory changes, and neurodegeneration to the brain due to *increased* lipid peroxidation or mitochondrial superoxide DNA damage in striatal cells due to dopamine oxidation Enhanced pyramidal activity in adult rats *Decreased* synaptic transmission and neuronal excitability in juvenile rats Loss of astrocytes and neurons with *increased* levels of cytokines and neurotrophins in juvenile ratsWistar rats [65,66]Sprague-Dawley rats [1]Wistar rats[60]Acute*Decreased* sensitivity to a given reward, *Decreased* habituation to a familiar environment and *Increased* depressive-like behavior*Increased* cross-sensitization suggests *increased* risk of future drug abuse *Increased* cocaine self-administration by rewarding effects and sensitivity of a given drugSprague-Dawley rats [61]Wistar rats [62] Sprague-Dawley rats [63] Sprague-Dawley rats [64] Neuroprotective effects observed via the reduction in cell damage and *decreased* apoptosis in brain tissueSprague-Dawley rats [68]


### 4.2. MP Oral Gavage

The oral gavage route of administration has been the most widely used and preferred technique for MP oral dosing in experimental studies. Oral gavage better mimics the oral consumption and metabolism of MP in humans [69]. The oral gavage method involves using a properly fitted tube or a gavage needle that is placed in an animal’s mouth and passed into the esophagus (See Table 2) [70]. Oral gavage is used for precise, accurate dosing and quick delivery of a drug [50]. The most commonly used dosage of MP through the oral gavage method is 2.5 mg/kg, mainly due to its calming effects shown in “ADHD rats”. However, this dose contrarily causes an excitatory response in wild type rats [71]. Doses used in other studies include 0.5–5 mg/kg, with 5 mg/kg on the higher end of the spectrum and commonly used to mirror illicit use of MP [72]. Other studies that used the oral gavage method to administer MP doses of 1, 10, and 50 mg/kg have shown adverse clinical observations including changes in body weight, pathology, and organ weight [73]. Researchers using the oral gavage method to administer drugs need to be extra careful since complications to the animals can occur. When poorly executed, this method can cause serious health concerns to the rat and could potentially cause aspiration and pulmonary injury to the animal [14]. Therefore, it is important to select appropriate tubing size and to handle the animal with extra care to minimize any sort of discomfort [50]. The goal of the oral gavage is to reach the peak MP plasma level within 0.5–1 h after administration [74]. The delivery of the MP using an oral gavage is shown to be more accurate and has a higher absorption rate than that of the dietary route [73].

#### 4.2.1. Effects of Gavage MP on Behavior

The oral gavage method is attributed to increased animal stress because this is a technique that requires stiff, manual restraint of rodents. This type of restraint is shown to increase plasma corticosterone levels in both rats and mice [70]. Therefore, it is important for researchers who perform oral gavage to undergo training prior to administration to prevent or minimize adverse events associated with the technique [50]. Studies have reported that there have been no negative behavioral effects of chronic MP exposure via oral gavage in healthy rodent subjects [75]. In another study [76], prolonged MP exposure in adult rats is shown to produce depressive-like behaviors linked to reduction in cell proliferation in the hippocampus. A study by Ponchio et al. [77] investigated if maternal behavior is affected by oral MP exposure via gavage. Maternal behavior, such as maternal care of pups, is impaired in female mice when MP has been administered repeatedly over the course of lactation (three days). Maternal behavior impairment is said to be a consequence of reduction in motivational behavior in female mice during the course of MP exposure. Other than that, impairment of maternal care is shown to increase anxiety-like behavior of pups when they reach adulthood [77]. MP administered acutely via oral gavage is shown to alleviate anxiety in Kv1.3 knockout mice that have heightened anxiety levels, but has no effect on wild-type mice (Refer to Table 2) [78]. The oral gavage method is used by researchers to simulate the clinical use of MP when studying the consequences of either short- or long-term MP exposure, and locomotor sensitization is studied to identify the abuse liability of a drug. Adolescent rats treated with chronic repeated MP via gavage show no development of locomotor sensitization, which could suggest that there is no abuse liability associated with psychotherapeutic treatment of MP [79].

#### 4.2.2. Effects of Gavage MP on Brain Function and Neurochemistry

The optimal dosage for administered MP through gavage is 5 mg/kg for all animal species [50]. Studies have shown that MP given via oral gavage increases extracellular DA levels in the brain, with no differences in dopaminergic responses as seen with other methods of administration [51]. In healthy animal subjects, there are limited neurotrophic consequences of chronic MP exposure via gavage. Striatal function activity and DA-mediated activated responses are seemingly unaffected by MP exposure when observed with MRI in the brains of healthy rodents (See Table 2) [75]. There is also no evidence of histopathological changes seen in tissues of the frontal cortex, striatum and hippocampus of adult male rats when treated with MP via oral gavage, which are brain regions where MP is believed to exert its clinical therapeutic effects in humans [69]. Some studies demonstrated that chronic exposure to psychostimulants, such as MP, decreases hippocampal neurogenesis but not neuronal differentiation in the hippocampus which is the brain region associated with addictive processes such as relapse and drug taking [76,80].


jpm-13-00574-t002_Table 2Table 2Summary of behavioral and neurochemical effects of gavage MP.Duration of MP ExposureBehavioral EffectsModel Used/ReferencesNeurochemical EffectsModel Used/ReferencesChronic*Decreased* animal stress Depressive-like behavior linked to *decreases* in hippocampal cell proliferation No evidence of changes in locomotor sensitization in adolescent ratsC57Bl/6J mice [70] Wistar rats [76]*Increased* plasma corticosterone *Increased* dopamine levels in the brain *Decreases* hippocampal neurogenesisC57Bl/6J mice [70]Sprague-Dawley rats [51]Wistar rats [76,80] Acute*Increases* animal stress Impairment of maternal behavior in female mice can produce pups with*Increases* anxiety-like behavior when they reach adulthood. Alleviates anxiety in Kv1.3 knockout miceC57Bl/6J mice [70] Inbred BALB C mice [77]Super-Smeller, Kv1.3 Knockout mice [78]*Increases* plasma corticosteroneC57Bl/6J mice [70]


### 4.3. MP Oral Voluntary Drinking

Rat physiology differs significantly from human. Rats typically have a faster metabolism [14], which will automatically decrease the half-life of any drug. Using a two-bottle regimen to administer MP over an 8 h period can potentially compensate for physiological differences between both species. A lower-dose administration for the first hour (bottle 1) and a higher dose administered for the other seven hours (bottle 2) allows for a corresponding method of exposure as oral MP consumption in humans. Each bottle given to the rats is adjusted daily to deliver a consistent concentration of MP [14]. Specific doses used included 4, 20 and 30 mg/kg for the first bottle and 10, 30, and 60 mg/kg for the second bottle [14]. Using this method of exposure of MP allows for a delivery method that is similarly attained in clinical administration while maintaining independence from fluid consumption [81]. Plasma levels obtained from the rats demonstrated that the two-bottle paradigm induced a 30 ng/mL plasma concentration [14]. This plasma concentration closely resembles those used in a clinical setting, 8–40 ng/mL [82].

#### 4.3.1. Behavioral Effects of Chronic Oral MP Treatment

Behavioral effects using a two-bottle paradigm can be measured using a multitude of tests. These tests include open field locomotor activity, elevated plus maze (EPM), forced swim test (FST), circadian activity, locomotor activity, sucrose preference, novel object, and cocaine CPP. Chronic exposure to MP can lead to an increase in anxiety, depression, and sensitization.


i.Open Field Locomotor Activity


Open field locomotor activity is a test that is commonly performed during the rat’s dark cycle and used to test for anxiety in the animal model. Sensitization is also examined throughout this testing and typically measured by monitoring the rat’s floor plane (FP) moves (total number of start to stop movements), FP distance traveled, and FP velocity [14]. Results from rats using a two-bottle exposure method to MP showed that the FP moves, velocity, and distance traveled were significantly increased during weeks of treatment in comparison to the vehicle [14,83]. IP injection of MP also significantly increased distanced traveled and velocity [54]. Increases in locomotor activity were found in both low dose- and high dose-treated females but only high dose-treated males, supporting the idea that there is greater sensitivity to psychostimulants in females than in males [84,85]. It has also been found that MP-induced increases in locomotor activity remain absent after 24 h of abstinence from MP, indicating that MP-induced increases in locomotor activity remain reversible following abstinence from MP [81]. It is also shown that in the dark cycle phase of circadian testing, MP induced an increase in locomotion [53]. Females were also found to be more sensitive to the locomotor-activating effects of MP and showed increases in exploratory behavior [86].

Another test that has been used to measure the effect of MP exposure employing a two-bottle paradigm is locomotor activity. Locomotor activity is measured in an enclosure that closely mirrors the rat’s home cage: researcher’s provide habituation time to allow for acclamation to a new environment. A significant effect of the MP two-bottle paradigm was observed on locomotor activity in comparison to the vehicle [14]. Consistent results were observed in another study (See Table 3) [81]. This study used a similar process as the two-bottle regimen previously mentioned. As stated before, doses ranging from 2 to 5 mg/kg resembled clinical use and demonstrated no significant increase in locomotor activity, whereas illicit use of MP (10–20 mg/kg) showed a significant increase in locomotor activity [87]. It has been shown that IP injection exposure demonstrates quicker effects of MP than voluntary oral gavage. Studies have shown that an oral exposure typically requires the dose to be above 5–10 mg/kg to begin showing a significant increase in locomotor activity [51]. Similarly, a study by Martin et al., 2018 assessed open field activity during the MP treatment phase, as well as the abstinence phase and found that MP increased locomotor activity in both males and females (59.1% and 95.9%, respectively), by observing increased distance travelled in an open field [85].


ii.Sleep/Circadian Activity


Circadian activity is an additional way to measure the success of the two-bottle regimen used for exposure to MP. This activity is measured by beam breaks, meaning the rat moves through the beam and breaks the path [81]. Rats exposed to MP resulted in a significantly higher activity measurement than the vehicle [14,85]. In male and female rats, more wake activity was seen during the dark cycle in MP-treated rats than in water-treated rats [86]. Female rats were more active than male rats during the dark cycle, but there was no difference experienced during the light cycle [86]. Once again similar results were observed in other studies using a two-bottle paradigm [81]. Like most testing that measures circadian activity, IP injections of MP have shown to increase activity. This idea also correlates to another study where MP increased circadian activity, with a greater increase in female rats compared to male rats [88]. Along with this, it has been demonstrated that there is a significant negative correlation between body weight and dark cycle circadian activity, and a positive correlation between dark cycle circadian activity and food intake [86]. MP-induced increases in locomotion were recorded during the dark cycle phase of circadian testing, during the hours when the subjects consumed MP, but not during the light cycle phase when they were deprived [83,85]. In terms of sleep, one study states that MP had no significant effect on multiple sleep parameters, but also states that this statistic may be due to their dosing schedule [14].


iii.Anxiety-Elevated Plus Maze (EPM)


Typically used to measure anxiety, the elevated plus maze (EPM) is another test used to measure behavioral effects of MP using a two-bottle paradigm. During this test, a rat is placed in the center of the EPM and their time spent in the open arms versus the closed arms is compared [81]. If a rat spends more time in the enclosed arms, this is a good indication that the rat is anxious. MP-exposed rats were shown to spend more time in the open arms than water-treated rats. This represents a less anxious animal [81]. Commonly, when exposed to MP using IP injections, the animal spent significantly more time in open arms than in closed arms [54]. This study agrees with the previous, and further shows that MP treatment decreases anxiety in the open field using the EPM [86]. Another study achieved the same results, that high-dose rats spent more time in the open arm than low-dose rats [85]. This study also revealed a sex-dependent anxiolytic effect of MP, with a greater proportion of time spent in the open arms by females [85]. This idea of anxiety relinquishment when treated with MP is used to look into other cognitive aspects and discrimination between “safe” and “unsafe” behaviors [14]. As a result of biological differences, male rats experienced less anxiety than female rats despite exposure to MP or non-exposure to MP [86]. This difference can be due to the differences in social stimuli used between male and female rats, with males being socialized with young rats to avoid aggression and female rats exposed to age and weight matched rats in the study [86]. It also remained clear that there is a significant difference between the high-dose MP group and water/low dose-treated groups throughout several of the studies [83].


iv.Depression-Forced Swim Test (FST)


Forced swim test (FST) was another form used to measure the effects of MP using a two-bottle paradigm. This test is commonly used to test for depressive-like behavioral effects in rats [81]. A rat demonstrates depressive-like behaviors by becoming immobile once placed in the water. Rats treated with MP displayed a greater latency to immobility when compared to water-treated rats [81]. Rats were less hesitant to swim and thus showed greater signs of depressive-like behaviors when treated with MP. This pattern/behavior is seen using the two-bottle paradigm. Using IP injections, no effect was observed when measuring latency to immobility [89]. The high-dose rats were also found to have a longer latency to immobility than the low dose- and water-treated groups [85]. It is clear that MP has an antidepressant-like effect on treated rats during the FST, particularly in high-dose groups, for both sexes [85]. Additionally, it was also found that rats treated with high doses of MP face a decreased time spent immobile in the FST as well [81]. Another study agreed with the significance of a high dose of MP in the FST and that the higher dose group spent less time in immobility. It also mentions that there is no significant difference between treatment groups during stages of abstinence [83].


v.Memory-Novel Object Recognition Memory


Novel object recognition is used to assess the short-term memory of a subject through the tendency of rodents to spend more time on a novel object than one that is familiar. Using a two-bottle paradigm, rats are placed in an open-field area with two objects in opposite corners. Animals are tested on the ability to differentiate between a novel object and a familiar one to assess short-term memory. Using an IP injection MP exposure method, MP improved cognitive function [52]. Cognitive malfunction refers to one’s ability to not remember. The rat’s ability to recognize familiar objects decreased following MP injection. It was also shown that chronic MP treatment had no effect on the degree of explanation of the familiar or novel object [86]. Other studies showed disrupted novel object exploration as a result of altered recognition memory [86]. One study shows that sessions conducted during the treatment phase did not have a significant effect on sex or interactions. However, during the abstinence sessions, significant effects of sex were shown. There were still no significant effects of interactions or time during this study [85]. Additionally, other studies reference that MP impairs performance in novel object recognition (Carias et al., 2018).


vi.Cocaine-Conditioned Place Preference


Cocaine-conditioned place preference (CPP) is used to measure behavioral effects such as reward and disinterest. CPP is run by having an operant box with three chambers. Two chambers have a different pattern on the walls, most commonly polka dots and stripes, and a center chamber with no pattern. Rats are injected with the drug of interest or saline (control). Following the injection, rats are placed in the center chamber and allowed to roam amongst the operant boxes. A video is taken from a bird’s eye view. The footage is later reviewed and wherever the rat spent the most time in is considered the “preferred” side. When rats are injected with the drug of interest, they are placed in the “non-preferred” side. When they are injected with saline, they are placed on their “preferred” side. For the next 8 days, rats alternated between drug and saline IP injections. On the last day, test day, the rats are not injected, and they are allowed to roam freely between chambers. During test day, researchers are looking for the rats to spend more time in the “non-preferred” side, demonstrating the drug-seeking behavior. A two-bottle paradigm model did not demonstrate any significant effect on cocaine CPP [90]. Other studies using the two-bottle paradigm have shown similar results: MP oral exposure does not exacerbate cocaine CPP [71]. However, IP injection of MP induced CPP when delivered at a 5 mg/kg dose [89]. Another study used IP injection exposure of MP and cocaine, using doses of 2/5 and 10/20 mg/k, respectively [63]. A higher dose of cocaine demonstrated a longer time spent in the drug paired chamber, non-preferred [63]. Other studies have contradicted this finding. Rats exposed to MP through an IP injection, 2.5 mg/kg for 25 days, and then later injected with cocaine (1.0, 5.0, 10.0 and 20.0 mg/kg of cocaine) showed no significant effect of MP exposure to cocaine CPP. There is no cohesive trend of MP exposure promoting cocaine CPP: more studies need to be done to generate a solid understanding of this behavior.


jpm-13-00574-t003_Table 3Table 3Comparison of behavioral effects using injection, oral gavage, and two-bottle paradigm.Route of MP AdministrationBehavioral EffectsModel Used/References
**Injected**
*Decreased* hyperactive behavior *Decreased* self-administration and reinstatement of drug-seeking behavior *Decreased* drug sensitization and tolerance in exploratory and object recognition memory Impaired spatial and working memory results in *Decreased* sensitivity to reward stimuli *Increased* locomotor activity compared to gavage administration *Increased* depressive and anxiety-like behavior *Decrease* in body weightNaples high-excitability rats [58]Spotaneously hypertensive rats [59]Wistar rats [60]Wistar rats [57]Sprague-Dawley rats [51]
**Oral Gavage**
*Decreased* animal stress Depressive-like behavior linked to *decreases* in hippocampal cell proliferation No evidence of changes in locomotor sensitization in adolescent ratsC57Bl/6J mice [70] Wistar rats [76]
**Two-Bottle Paradigm**
*Increased* locomotor activity *Increased* circadian activity*No effect* on sleep*Decrease* in anxiety in EPM*Increase* in latency to immobility during FST*No effect* on cocaine preference placement test*Increase* in food intake*Decrease* in body weightSprague-Dawley rats [81]Sprague-Dawley rats [90]Sprague-Dawley rats [14]Sprague-Dawley rats [85]


#### 4.3.2. Developmental Effects of Chronic Oral MP Exposure

Development can be affected when a subject is chronically exposed to MP. A rat’s development is measured through food intake, body weight, and skeletal growth.


i.Food Intake


Food intake is typically measured weekly, at a minimum. Using a two-bottle paradigm, rats were treated with MP and food consumption was monitored daily. Rats exposed to MP consumed more food than the vehicle group [85]. Similarly, IP injected MP rats demonstrated a greater food intake when compared to the control group. Other studies that used an oral route for MP exposure (2.0, 5.0, and 8.0 mL/kg) increased food intake. On the contrary, other studies have shown that higher doses of MP, 100 mg/kg, significantly decreased food consumption [73]. Reflecting back on previous papers, 100 mg/kg is a higher dose on the spectrum and would reflect illicit use in a clinical study. Some studies show that food intake is independent of growth regression, and that while food intake and growth regression increase at the same time, they are not factors independent of one another [88]. Weight gains were also shown to decrease while food intake increased [88]. Additionally, it is shown that high-dosed male and female rats tended to consume less food than water-treated male rats towards the beginning of their initial dosing, increasing their food intake after the treatment had extended over time [85]. Other studies also show this increasing food pattern over time in both males and females [86]. Average weekly food intake decreased by 10% in MP high-dosed rats and by 7% less than water-treated rats in MP low-dosed rats during the first 1–5 weeks of treatment [91], as well as demonstrating a significant drug/time interaction effect on food intake in both low- and high-dose rats of both sexes, but with an emphasis on female rats [14]. It is also possible that high dose-treated rats consume more fluid as compensation for suppressed fluid intake during MP treatment, showing that a prolonged abstinence period from MP treatment can reverse the behavioral and developmental effects of chronic MP usage over time [81]. It is also suggested that food intake has a significant correlation with dark cycle circadian activity, and no direct correlation between food intake and body weight overall [86].


ii.Body Weight


Monitoring body weight is another way to measure the developmental effect of exposure to MP using a two-bottle paradigm. Throughout treatment, MP-exposed rats tend to weigh less than water-treated rats [85]. As the rats grow and age, body weight increases but at a rate significantly less than that of the water-treated rats. This may be due to the fact that stimulants suppress appetite, which would result in body weight loss [90]. Body weights of MP oral treated rats in week 1 of a study were 11% lower than water control rats, progressing to a 13% deficit at the end of treatment (Week 13), with a weekly average of a 7% deficit over the course of the treatment [91]. Body weight was also reduced when the rat was exposed to MP via IP injection, 5 mg/kg [92]. Additionally, a study showed that high-dose MP treatment reduced body weight compared to the water treatment group throughout the total time of treatment. However, this effect took longer in females, starting at week 4 to show reduced body weight [86]. These results align with previous studies, which also show that treatment of MP, in high and low doses, attributes to reduced body weights over time [14,83]. Even further, when high-dose MP exposure is paired with intermittent abstinence periods, body weight is decreased and food consumption decreases sporadically [83]. Some of these studies conclude that when non-ADHD patients receive chronic treatment of MP, significant physiological effects (such as a reduction in body weight) can occur drastically over time [81]. Behaviorally, it was found that there was no correlation between body weight and food intake, yet there was a significant negative correlation between body weight and dark cycle circadian activity [86]. Though the correlation between body weight and food intake did not exist in rats that were euthanized immediately following treatment, it was found that a positive correlation ensued in rats euthanized following a period of abstinence [86]. Some studies using the oral gavage method did not demonstrate a significant decrease in body weight from MP exposure [73].


iii.Skeletal Effects


Observation of bone health is important because it allows one to see if the drug being administered is causing any growth deficits in the subject. Densitometry and bone sizes are both used to measure bone health. There is a decrease in anterior–posterior (AP) bone diameter in rats who consume chronic MP [91], signifying inadequacy in growth. Densitometry is often used to assess a weakening of the bones. A degradation in bone mineral density (BMD) and bone mineral content (BMC) was observed in rats that consumed MP through a two-bottle paradigm [91]. A slight increase in the density of class 2 and core spine was observed in other studies when the animal model was exposed to MP via 15 mg/kg IP injection [93]. A two-bottle paradigm, using the doses 4/30 mg/kg for the first hour and 10/60 mg/kg for the remainder of the day demonstrated a significant increase in osteoclasts on the surface of the cortical bone and TRAP volume fraction [88]. Other studies demonstrate chronic exposure to MP doses of 4/10 mg/kg (first bottle) and 30/60 mg/kg (second bottle) significantly decreases skeletal growth and mineralization [91]. The bone biomechanical properties of ultimate force and stiffness both showed positive correlations with body weights in MP-low rats [91]. AP diameter on ultimate force, F(1610) = 16, *p* < 0.001 and stiffness F(1,2930) = 8.5, *p* < 0.0 with small effects and an 8.6% variance for ultimate force and a 0.3% variance for stiffness accounted for by AP diameter [91]. A clinical study has shown that stress fractures are reduced when an individual is orally exposed to MP [94]. Studies have reported reduction in height gain in subjects that use MP [95]. In the multimodal treatment study of ADHD (MTA), 7–9-year-old children affected by MP showed growth suppression of 0.9 cm/yr over a 14-month period, and an additional 1.04 cm/yr in the 10 month follow up period. Stronger suppression of growth was seen in the Preschool ADHD Treatment study (PATS) where growth suppression spanned 1.38 cm/yr in 4–5-year-old children [96]. Some studies do not express a difference in growth in MP oral-treated subjects, so these findings are highly debated.

In a more recent study, the effects of combined MP and fluoxetine (FLX) treatment on skeletal development were evaluated [97]. Using the dual-bottle approach and 4 groups of rats exposed to water, MP, FLX, and MP + FLX, results showed that MP + FLX-treated rats had significantly shorter (~12%) and narrower femora and tibiae (~10%) compared to other groups. In addition, these rats also displayed shorter (26–35%), disorganized tibial growth plates. Similarly, most trabecular and cortical microstructural parameters were also reduced in MP + FLX rats; proximal tibia reductions of 47% for total volume (TV), 86% for bone volume (BV), 74% for BV/TV, 68% for trabecular number, and 25% in trabecular thickness that was concomitant with increases of 44% for trabecular spacing and 3% for volumetric bone mineral density for MP + FLX compared to water. Similar reductions were observed with femoral midshaft cortical bone; 29% for cortical volume, 30% for periosteal volume, 30% for endocortical volume, and 51% for polar moment of inertia, as well as increases of 17% for cortical thickness and 2% for tissue mineral density (define this for MP + FLX compared to water. Lastly, MP + FLX femora were biomechanically weaker; there was a reduction in ultimate force (14%) in MP + FLX compared to water [97]. Interestingly, these microstructural and biomechanical effects of MP + FLX were eliminated after adjustment for body weight, but the detrimental effects on growth plate morphology were unchanged. Based on these data, the authors concluded that while the adverse microstructural and biomechanical effects of MP + FLX are predominantly attributable to reductions in body weight, their findings warrant additional research into the effects of these drugs on weight gain and skeletal development, and that this should be considered by physicians treating children and adolescents with ADHD (Table 4) [97].

#### 4.3.3. Neurochemical Effects of Chronic MP

MP increases extracellular DA by blocking its reuptake. Most stimulants work in a similar manner [98]. The most commonly affected areas in the brain are the PFC, striatum, and nucleus accumbens [98]. The PFC is most commonly associated with cognitive behavior, decision making, and moderating social behavior. The striatum is highly involved in motivation and reward. Finally, the nucleus accumbens plays a key role in the reward circuitry and is mainly affected by the neurotransmitter DA [86].


i.DA Receptors


Dopamine D1 receptors are commonly found in the striatum and nucleus accumbens. They assist in the development of neurons when they are bound to DA. While receiving MP through a two-bottle paradigm, the low-dose (LD) group received 4 mg/kg in the first hour, and for the remaining 7 h, received 10 mg/kg. The high-dose (HD) group received 30 and 60 mg/kg, respectively. This method of exposure resulted in a significant increase in DA D1 receptors in regions of interest such as the middle dorsolateral, dorsomedial, ventrolateral, and ventromedial caudate putamen [86]. It was also found that MP exposure using the two-bottle paradigm had no significant effect on DA D2 receptor binding between treatment groups [86]. The effects of chronic oral MP were further examined in the Thanos et al. (2007) study. MP was given orally at doses of 1 and 2 mg/kg and microPET imaging showed that the 1 mg/kg dosage group exhibited significantly less DA D2 receptor binding compared to the 2 mg/kg dose group [40]. Chronic MP use in rats altered DA 2 receptor (D2R) expression levels in the striatum, varying in either increased or decreased expression depending on the length of treatment and the age of rats during drug administration [40].

The effects of MP on DA receptors are shown in Table 5. PET imaging research has also found MP to preferentially target the ventral striatum, a region with high amounts of DAT, which may lead to decreased levels of impulsivity in animals [99]. DA D2 receptors are known to be inhibitory to DA release, allowing for the regulation of DA within the synaptic cleft. Studies have shown that using the two-bottle paradigm, HD MP exposure, as compared to LD, significantly increases DA transporter (DAT) binding in subregions of the basal ganglia such as the caudate nucleus and putamen [86]. The DAT works by removing DA from the synaptic cleft. Removal of DA from the synaptic cleft is important in maintaining tolerance to certain drugs. Constant stimulation of MP will lead to a downregulation of DA receptors, along with developmental and behavioral effects such as attenuated body weight and increased dark cycle circadian locomotor activity (See Table 5) [14].


ii.NMDA Glutamate Receptors


While MP has been noted to act predominantly on DA and NE, it is possible that it may also have an effect on the glutamatergic system to enhance attention and memory as well since these systems are connected by signaling [39]. N-Methyl-D-Aspartic acid (NMDA) receptors are glutamate influenced channels that allow for regulation in memory [100]. Previous studies have demonstrated reduced production of NMDA receptors after one IP injection of MP. Acute doses of MP treatment significantly decreased protein levels of NMDA glutamate receptors in the PFC of young rats [42]. In a study done by Jalloh et al. (2021), the two-bottle paradigm was used in two groups of rats: LD 4 mg/kg for one hour and 10 mg/kg for the remanding 7 h, and HD received 30/60 mg/kg respectively. This method of exposure resulted in decreased binding of MK801, an NMDA antagonist, ultimately increasing the concentration of NMDA in the hippocampus, primary motor cortex, auditory cortex, amygdala, thalamus and dorsal striatum (Table 5) [101]. These findings are shown in Table 5. Altered NMDA receptor expression from MP treatment may lead to effects on fear conditioning, memory and drug-seeking behavior. Interestingly, the effects on NMDA receptor binding were reversible following a 4-week abstinence period. This reversibility of chronic MP treatment has been seen behaviorally and has been observed with respect to DA receptor binding as well.


iii.CB1 Cannabinoid Receptors


Research has found that chronic oral MP use may also have dose-dependent effects on the endocannabinoid system of the brain. CB1 is the most abundant receptor in the mammalian brain. Using a two-bottle chronic exposure paradigm of MP (LD (4/10 mg/kg) or HD (30/60 mg/kg), followed by a four-week abstinence period [102], researchers were able to demonstrate an effect on the brain’s endocannabinoid signaling, particularly the CB1 cannabinoid receptor. CB1 receptors inhibit neurotransmitter release. In this study, HD MP exposure demonstrated a significant increase in [3H] SR141716A binding in several regions of the basal ganglia, PFC, ventral tegmental area, and the hindlimb region of the somatosensory cortex. [3H] SR141716A acts as an antagonist for cannabinoid CB1 receptors. Following a 4-week abstinence period, LD MP exposure resulted in a significant decrease in [3H] SR141716A binding, while abstinence from HD MP resulted in an increase in CB1 receptor level binding (Table 5) [102]. These findings are summed up in Table 5.


jpm-13-00574-t005_Table 5Table 5Summary of the neurochemical effects of chronic oral MP.ReceptorBrain Regions of InterestFindingsModel Used/ReferencesDopamine
Basal gangliaOlfactory tubercle

HD MP rats had *increased* DAT binding in the caudate putamen compared to water-treated rats and LD MP ratsHD MP rats had increased D1R-like binding in the olfactory tubercle and caudate putamenLD MP had no effect on any binding measured in any region assessedNeither dose of MP influenced D2R-like binding
Sprague-Dawley rats [86]Dopamine
Striatum

*Decreases* in striatal D2R availability at 2 months of tx*Increases* at 8 months of MP tx
Sprague-Dawley rats [40]NMDA
HippocampusPrimary motor cortexAuditory cortexAmygdalaThalamusDorsal striatum

Chronic HD MP rats showed significant decreases in NMDA receptor bindingThe effects were reversible following a 4-week abstinence period.
Sprague-Dawley rats [101]CB1 Cannabinoid
Basal gangliaHindlimb region of the somatosensory cortex

Chronic HD MP rats showed *decreases* [3H] SR141716A binding in the caudate putamen regions compared to the LD MP groupAbstinence from HD MP resulted in an *increase* in CB1 receptor levelsAbstinence from LD MP resulted in a *decrease* in CB1 levels
Sprague-Dawley rats [102]


#### 4.3.4. Effects of Chronic Oral MP on Brain Function and Structure

Additional studies have also found age-dependent effects of chronic oral MP use on brain function. A study conducted by van der Marel and colleagues [76] investigated the effects of chronic oral MP use on recognition memory, functional connectivity networks, hippocampal shape, and markers for hippocampal neurogenesis and proliferation in both adult and adolescent rats. Male rats, both adult and adolescent, were given oral MP for a period of 21 days, then subjected to behavioral tests and both MRI and fMRI imaging. The research showed increased levels of neurogenesis in adolescent-treated rats but decreases in cell proliferation in adult-treated rats. Additionally, deformations were found near the ventral parahippocampal regions in adolescent rats in areas involved in recognition memory, while these changes were not seen in adult-treated rats. However, both adult and adolescent rats did show functional network changes in the ventral hippocampal regions of the brain.


i.MP Effects on Brain Function


A recent study [103] aimed to analyze the effects of chronic MP use on brain function in adolescent rats. Using 18F-FDG along with PET neuroimaging, it analyzed whether there were changes in BGluM, and where in the brain those changes occurred. Additionally, the study aimed to address whether or not differences would reverse after abstinence from the drug. The experiment assigned 36 male, 4-week-old Sprague Dawley rats randomly into three groups: the control group, LD MP group, and HD MP group. The LD MP rats received 4 mg/kg MP for the first hour and 10 mg/kg MP for the remaining 7 h of the 8 h treatment time while the HD MP rats received 30 mg/kg MP for the first hour and 60 mg/kg MP for the remaining 7 h. Rats were administered the treatment for a schedule of 5 days on treatment and 2 days off, for a period of 13 weeks. At the end of the 13-week treatment period, rats underwent FDG-PET scans. Afterwards, rats were abstained from MP administration for a period of 1 week, underwent a second PET scan, then abstained for an additional 3 weeks for a total of 4 weeks abstinence from the drug, followed by a third PET scan. Results indicated higher levels of activation in the brain after LD MP treatment in the hippocampus, subiculum, and simple lobule; after HD MP treatment higher activation was found in the sensorimotor cortex, primary auditory cortex, ectorhinal cortex, temporal association cortex, and inferior colliculus. The brain also exhibited regions with decreased BGluM in HD MP rats compared to LD MP rats in the cingulate cortex. This indicates that changes in BGluM in the rat brain are dose dependent. Additionally, after abstinence, a complete reversal in effects were found in the LD MP groups at the end of the 4-week abstinence period and a reversal in effects were found initially during the 1-week abstinence from HD MP groups, but that changed to show activation at the end of the 4-week abstinence period in HD MP rats in the hippocampus and retro splenial cortex. Thus, contrasting results were observed depending on dosage administered and length of abstinence period.

Another recent study [104], aimed similarly to analyze the effects of chronic MP use on brain function. Combining 18F-FDG with PET neuroimaging, researchers analyzed changes in BGluM in the brains of adolescent rats after chronic MP use. Additionally, the study analyzed reversibility of the effects after abstinence from the drug. The same methods [103] were utilized. Briefly, 4-week-old rats were randomly separated into three groups: the control group, LD MP group, and HD MP group. The LD MP rats received 4 mg/kg MP for the first hour and 10 mg/kg MP for the remaining 7 h of the 8 h treatment time while the HD MP rats received 30 mg/kg MP for the first hour and 60 mg/kg MP for the remaining 7 h. This experiment differed, however, in the schedule of MP administration: rats were administered the treatment for a schedule of 3 weeks on treatment and 1 week off treatment, for a period of 13 weeks. At the end of the 13-week treatment period, rat FDG-PET scans were taken. Then, rats were abstained from MP administration for a period of 1 week, completed a second PET scan, then abstained for an additional 3 weeks, for a total of 4 weeks of abstinence from MP. This was followed by a third PET scan. Unlike previous findings [103], this study did not find changes in BGluM between the MP-administered and control groups. However, changes in BGluM were observed after 1-week abstinence between the HD MP group and control group in areas such as the cerebellum, sagulum, lateral lemnicus, inferior colliculus, and entorhinal field. Additionally, after 4 weeks of abstinence, even more areas in the brain seemed to be activated when comparing the HD MP and control groups, such as the cerebellum, mesencephalic reticular formation, inferior colliculus, lateral lemniscus, trigeminal nucleus, trigeminal nerve, inferior olive, and other areas. This study not only shows dose-dependent differences between results, but may also indicate that changes in BGluM and effects on reversibility may be altered depending on the schedule of drug use.


ii.MP Effects on Brain Structure


The effect of chronic MP use on neuromorphology has been previously examined [105]. Briefly, adolescent male rats were treated with either water or a low dose of MP (4 mg/kg MP for the first hour and 10 mg/kg MP for the remaining 7 h of the 8 h treatment time) or a high-dose MP (30 mg/kg MP for the first hour and 60 mg/kg MP for the remaining 7 h). These concentrations were chosen to produce plasma MP concentrations within the clinical pharmacokinetic range as previously described [14]. After 3 months of daily treatment, some rats underwent 9.4T structural MRI. Imaging results concluded that chronic oral MP use with either dose decreased the brain volume of the posterior white matter in areas such as the cingulum bundle, dorsal hippocampal commissure, and the external capsule [105]. More specifically, in the corpus callosum/external capsule, chronic MP results in a 11–13% volume decrease (Table 6). The cingulum bundle is a group of white matter association fibers that connect retrosplenial and cingulate cortices with prefrontal, parietal, and limbic cortices, with the striatum, anterior thalamus, and pons. The dorsal hippocampal commissure contains interhemispheric fibers connecting to the entorhinal cortex, presubiculum, parasubiculum, and retrosplenial cortex. These structural neuroanatomical changes in white matter tracts are hypothesized to potentially lead to brain function changes, which may or may not be reversible.

## 5. Conclusions

The two-bottle drinking paradigm is a method of oral MP treatment that accurately mirrors the oral treatment observed in humans. This approach demonstrated that chronic oral MP produced a reduction in anxiety and depression-like behaviors, while increasing food consumption and decreasing bone density. In contrast, it decreased body weight. In addition, chronic (two-bottle drinking) oral MP treatment produced an increased expression of DA and NMDA receptors in the PFC, resulting in a stimulation of the PFC. Stimulation of the PFC allows for higher functions such as attention and motivation. These data support the utility and application of the two-bottle drinking MP paradigm as a key rodent model to studying MP or psychostimulant chronic effects in humans. The data presented support the use of this approach in studying the pharmacodynamics of MP in rodent models.

## Figures and Tables

**Table 4 jpm-13-00574-t004:** Summary of behavioral and developmental effects on oral MP.

Developmental Effects	Model Used/References	Behavioral Effects	Model Used/References
**Food intake:** Rats exposed to MP were prone to *increase in* food intake when compared to the control group.	Sprague-Dawley rats [85]	**Open field locomotor activity:** Rats exposed to MP have *increased* locomotor activity. Rats injected with MP effected faster than by MP Oral voluntary.	Sprague-Dawley rats [14] Sprague-Dawley rats [51]
**Body Weight:** Rats exposed to MP tend to have *decreased* body weight when compared to the control group. Body weights tend to *decrease* over time, with more significant *decreases* in the later weeks of treatment, espescially in females.	Sprague-Dawley rats [85] Sprague-Dawley rats [14]	**Sleep and Circadian:** Both male and female experienced more activity than water rats during the dark cycle. Females experienced more activity than male.	Sprague-Dawley rats [86]
**Skeletal Effects:** Rats exposed to MP demonstrated an *increase* in osteoclast formation on the surface of the cortical bone. Shown to experience a *decrease* in skeletal growth and mineralization. *Decrease* in stress fractures Some studies indicate growth suppression.	Sprague-Dawley rats [88] Sprague-Dawley rats [91] Human children [94]	**Anxiety:** Rats exposed to MP are shown to have *decreased* levels of anxiety. Male experience less anxiety than female rats due to biological differences.	Neonatal 6-hydroxydopamine mice [81] Sprague-Dawley rats [86]
		**Depression:** MP rats displayed a >latency to immobility compared to water-treated rats. Rats showed *increased* signs of depressive symptoms.	Sprague-Dawley rats [81]
		**Memory:** Some studies showed an *increase* in cognitive malfunction in rats exposed to MP. Other studies showed distruption in novel object exploration as a result in altered memory in rats exposed to MP.	Neonatal 6-hydroxydopamine mice [52] Sprague-Dawley rats [86]
		**Cocaine-Conditioned Place Preference:** A higher dose of cocaine demonstrates greater drug-seeking behavior. There is no cohesive trend of MP exposure promoting Cocaine CPP.	Sprague-Dawley rats [63]

**Table 6 jpm-13-00574-t006:** Summary of the BGluM changes following oral MP treatment. Areas of activation are indicated in red and areas of inhibition are indicated in blue.

	Arnavut et al., 2022 [105]	Richer et al., 2022 [104]
**Treatment** phase: LD MP > Control	None	Hippocampus and subiculum and simple lobule
**Treatment** phase: HD MP > Control	None	Sensorimotor cortex, primary auditory cortex, ectorhinal cortex, inferior colliculus, and temporal association cortex
**Treatment** phase: HD MP > LD MP	Primary and secondary visual cortex Ectorhinal cortex	Cingulate cortex and striatum
**Abstinence** phase: 1 week LD MP > Control	None	Medial orbital cortex, lateral hypothalamus, hippocampus, and subiculum
**Abstinence** phase: 1 week HD MP > Control	Superior cerebellar peduncle, tectospinal tract, sagulum nucleus and lateral lemniscus Inferior colliculus, caudomedial entorhinal field, pedunculopontine tegmental nucleus, 5th and 6c cerebellar lobule, cerebellar nucleus and cerebellar white matter	None
**Abstinence** phase: 1 week HD MP > LD MP	Sagulum nucleus, lateral lemniscus, external cortex of the inferior colliculus, caudomedial entorhinal field, spinal trigeminal nucleus, dorsomedial spinal trigeminal nucleus, parvicellular reticular nucleus	Medial orbital cortex, insular cortex, basal/lateral amygdaloid nucleus, dorsal endopiriform nucleus, lateral amygdaloid nucleus and pontine reticular nucleus
**Abstinence** phase: 4 weeks LD MP > Control	None	None
**Abstinence** phase: 4 weeks HD MP > Control	Mesencephalic reticular formation, lateral and medial lemniscus, microcellular tegmental nucleus, inferior colliculus Inferior olive, trigeminal nucleus and nerve trapezoid body, motor trigeminal nucleus and nerve, cerebellar white matter, crus 2 of the ansiform lobule, 8th cerebellar lobule, reticular nucleus, spinal trigeminal, and pyramidal tract	Hippocampus, retrosplenial cortex
**Abstinence** phase: 4 weeks HD MP > LD MP	Preoptic area, hypothalamic area, parasubiculum, inferior colliculus, retrosplenial dysgranular cortex, trigeminal nucleus, medioventral periolivary nucleus, trapezoid body, pyramidal tract, and gigantocellular reticular nucleus	Cuneiform nucleus
**Abstinent** phase 1 week LD MP > **Treatment** phase LD MP	None	Nucleus accumbens, striatum, lateral hypothalamus, hippocampus, solitary nucleus
**Treatment** phase HD MP < **Abstinent** phase 1 week HD MP	None	Ventral and medial orbital cortex
**Treatment** phase LD MP > **Abstinent** phase 4 weeks LD MP	None	Striatum, insular cortex
**Treatment** phase HD MP > **Abstinent** phase 4 weeks HD MP	None	Ventral and lateral orbital cortex & anterior olfactory nucleus

## Data Availability

Data is available upon request to the corresponding author.

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
