# Peer review of "Behavioral, Neurochemical and Developmental Effects of Chronic Oral Methylphenidate: A Review"

_jpm, 2023, doi:10.3390/jpm13040574_

Round 1
Reviewer 1 Report
Review entitle “Behavioral and Neurochemical and Developmental Effects of chronic Oral Methylphenidate: A Review”, I found this is very interesting review article for the science community, which discuss about the behavioral, neurochemical and developmental effects of Methylphenidate. Especially for clinically used drugs we need such type of review articles, wherein special focus should be on how drug administration routes affect the drug’s pharmacokinetics and pharmacodynamics as well as how an acute and chronic use will produce the behavioral, neurochemical and developmental side effects. I appreciate an idea behind the writing this review, however this review mainly discuss about preclinical studies, this will be an excellent review article for the clinician.
I have few minor suggestions that authors should consider to include in present review.
1. Table 1,2,3 and 4 need to be formatting again, I think instead of up & down arrows and special characters >, authors should directly write increase/decrease.
2. If it is possible for the authors then please include the mouse/rat stains used in that respective studies, you can add this information in the table only.
3. Authors should add one summary table for the present review, wherein they will show the comparative behavioral side effects of 2 bottle paradigm versus other route of administrations, table 1 and 2 is already discussing this for injected and oral gavage respectively but missing comparison with 2 bottle paradigm.
Author Response
Comments and Suggestions for Authors
Review entitle “Behavioral and Neurochemical and Developmental Effects of chronic Oral Methylphenidate: A Review”, I found this is very interesting review article for the science community, which discuss about the behavioral, neurochemical and developmental effects of Methylphenidate. Especially for clinically used drugs we need such type of review articles, wherein special focus should be on how drug administration routes affect the drug’s pharmacokinetics and pharmacodynamics as well as how an acute and chronic use will produce the behavioral, neurochemical and developmental side effects. I appreciate an idea behind the writing this review, however this review mainly discuss about preclinical studies, this will be an excellent review article for the clinician.
I have few minor suggestions that authors should consider to include in present review.
- Table 1,2,3 and 4 need to be formatting again, I think instead of up & down arrows and special characters >, authors should directly write increase/decrease.
- This was addressed with tables 1-5 on pages 45-49. All arrows have been changed to their respective indications.
- If it is possible for the authors then please include the mouse/rat stains used in that respective studies, you can add this information in the table only.
- All models were inserted by their respective references on tables 1-5 (pages 45-49).
- For table five the model was added in the caption.
- Authors should add one summary table for the present review, wherein they will show the comparative behavioral side effects of 2 bottle paradigm versus other route of administrations, table 1 and 2 is already discussing this for injected and oral gavage respectively but missing comparison with 2 bottle paradigm.
- Using table one and two an additional table (table 3, page 47) was made to compare the 3 routes of admission.
Reviewer 2 Report
JPM-2147754. Peer review. England, January 2023.
The review article by Senior and colleagues is focusing on oral Methylphenidate in pre-clinical studies, only using the two-bottle choice paradigm.
Line 34, the reference Urban 2012a is exactly the same as Urban 2012b. Please edit. Similarly, the references appearing in lines 957-963 are duplicates. Several other duplicates were also detected (969-976, 1024-1030, 1181-1189, 1208-1214, 1220-1228)
Grammar and style need editing, throughout. Please avoid inserting judgmental comments.
Lines 199-207. This part of the manuscript does not bring added value. Please remove. This part can be easily replaced with the different first- and second-pass metabolic differences between rodents and humans.
How did the authors select all of the included studies? Was a systematic review conducted?
The pharmacology of Methylphenidate is not discussed in details. This needs to be included in the manuscript.
Unfortunately, such a review does not bring new evidence in the current field, the latter already flooded with review articles on similar topics (Methylphenidate). Because of lack of originality, rejection should be recommended. Furthermore, such a review does not belong to the Special issue it is submitted to, nor the Section, as it is not remotly linked to these.
Author Response
Comments and Suggestions for Authors
JPM-2147754. Peer review. England, January 2023.
The review article by Senior and colleagues is focusing on oral Methylphenidate in pre-clinical studies, only using the two-bottle choice paradigm.
- Line 34, the reference Urban 2012a is exactly the same as Urban 2012b. Please edit. Similarly, the references appearing in lines 957-963 are duplicates. Several other duplicates were also detected (969-976, 1024-1030, 1181-1189, 1208-1214, 1220-1228)
- All duplicate references have been removed.
- Grammar and style need editing, throughout. Please avoid inserting judgmental comments.
- We have had the paper reviewed for grammar and style by the university writing center.
- Lines 199-207. This part of the manuscript does not bring added value. Please remove. This part can be easily replaced with the different first- and second-pass metabolic differences between rodents and humans.
- This has been removed.
- How did the authors select all of the included studies? Was a systematic review conducted? This was a systematic narrative review using PUBMED.
- The pharmacology of Methylphenidate is not discussed in details. This needs to be included in the manuscript.
- This has been addressed in the introduction (pages 3-6).
- Unfortunately, such a review does not bring new evidence in the current field, the latter already flooded with review articles on similar topics (Methylphenidate). Because of lack of originality, rejection should be recommended. Furthermore, such a review does not belong to the Special issue it is submitted to, nor the Section, as it is not remotely linked to these. We disagree with this comment as this paper summarizes important aspects of the literature that is completely missing. The present review thus provides a comprehensive review of the effects of oral MP which is one of the most prescribed psychiatric medication.
Reviewer 3 Report
Thank you for this interesting paper that compares routes of administration of methylphenidate (MP) in animal studies, i.e., intraperitoneal injections, subcutaneous injections and oral gavage, and which suggests the superiority of MP oral voluntary drinking. Indeed, a limitation of animal studies is that the route of administration of MP is not oral as is used clinically.
As nicely highlighted by this study, humans who are treated for ADHD receive MP orally, either in the immediate- or the extended-release formulation. In most animal studies, MP is administered intravenously, intraperitoneally, or subcutaneously.
The paper reviews thoroughly several studies that show that these routes of MP administration differ significantly from oral administration, specifically with respect to magnitude of and time to peak serum concentration, half-life, and rate of elimination, as well as absolute magnitude and time course of increases in extracellular dopamine and locomotor responses.
The paper is well written, albeit lengthy – I wonder if the Authors could synthesize the results of the studies that they review and present them more succinctly.
This is my only comment for the Authors.
The discussion is well balanced. There is not really much to criticize about this manuscript.
Author Response
We have reviewed the comments supplied by Reviewer 3 and are pleased that they concluded that, “There is not really much to criticize about this manuscript.” In fact, the only concern raised was in the comment that, “I wonder if the Authors could synthesize the results of the studies that they review and present them more succinctly.” We feel that we have reviewed the literature in as concise a manner as possible and believe that a briefer presentation of these studies would take away from the value of our review. Furthermore, we have provided five tables that present the results in a very succinct manner. As such, we do not feel that additional revisions to the manuscript to address this reviewer’s comments will improve out manuscript and respectfully are not making any additional revisions.
Of note, we did make significant revisions to address the concerns of the first two reviewers and this revised manuscript has already been uploaded. We hope that you will find the current revision acceptable for publication in the Journal of Personalized Medicine.